# A Combined Raman Spectroscopy and Atomic Force Microscopy System for In Situ and Real-Time Measures in Electrochemical Cells

**DOI:** 10.3390/ma16062239

**Published:** 2023-03-10

**Authors:** Gianlorenzo Bussetti, Marco Menegazzo, Sergei Mitko, Chiara Castiglioni, Matteo Tommasini, Andrea Lucotti, Luca Magagnin, Valeria Russo, Andrea Li Bassi, Martina Siena, Alberto Guadagnini, Samuele Grillo, Davide Del Curto, Lamberto Duò

**Affiliations:** 1Department of Physics, Politecnico di Milano, 20133 Milan, Italy; 2NT-MDT BV, Hoenderparkweg 96 b, 7335 GX Apeldoorn, The Netherlands; 3Department of Chemistry, Materials and Chemical Engineering, Politecnico di Milano, 20133 Milan, Italy; 4Department of Energy, Politecnico di Milano, 20133 Milan, Italy; 5Department of Civil and Environmental Engineering, Politecnico di Milano, 20133 Milan, Italy; 6Department of Electronics, Information and Bioengineering, Politecnico di Milano, 20133 Milan, Italy; 7Department of Architecture and Urban Studies, Politecnico di Milano, 20133 Milan, Italy

**Keywords:** solid–liquid interface, HOPG intercalation, in situ AFM, Raman spectroscopy

## Abstract

An innovative and versatile set-up for in situ and real time measures in an electrochemical cell is described. An original coupling between micro-Raman spectroscopy and atomic force microscopy enables one to collect data on opaque electrodes. This system allows for the correlation of topographic images with chemical maps during the charge exchange occurring in oxidation/reduction processes. The proposed set-up plays a crucial role when reactions, both reversible and non-reversible, are studied step by step during electrochemical reactions and/or when local chemical analysis is required.

## 1. Introduction

Scanning probe microscopy (SPM) techniques, such as scanning tunneling microscopy (STM) and atomic force microscopy (AFM), were developed in the 1980s [1,2]. Since their introduction in surface science, these were commonly perceived as revolutionary microscopy yielding something like “a lab in a tip” [2]. Two main drawbacks limit the powerful potential of SPM: (i) difficulty of application in liquid environments for, e.g., corrosion studies, and (ii) the chemical blindness of the technique. The major implication of the latter is that regions of a sample characterized by the same morphology while having different chemical properties appear virtually indistinguishable under the tip scan. To overcome the latter issue, two main strategies have been adopted: (1) tip functionalization, where specific atoms or molecules are bound to the tip apex to enhance sensitivity to the chemical properties of the surface of the sample [3]; and (2) coupling with spectroscopic techniques, such as reflectivity [4], IR [5], or Raman spectroscopy [6]. With reference to the latter, the experimental configuration most often adopted envisages placing the AFM tip on the top side of a transparent sample while the Raman laser source is focused on the back side of the specimen [7]. 

The investigation of surfaces immersed in liquid involves critical challenges. The first pioneering studies were performed by Hansma for both the STM and the AFM technique [8,9]. In this context, STM represents a markedly challenging set-up because the tip is biased with respect to the sample as the latter literally becomes an electrode when it is immersed in a conductive ionic liquid. This issue was overcome by Itaya in 1988 through the introduction of so-called electrochemical STM (EC-STM) [10]. EC-STM and EC-AFM were also coupled with spectroscopies [11,12]. Nonetheless, spectroscopic techniques have always been limited by the diffraction limit and their implementation has been thought to yield an average characterization of the surface where SPM can reach atomic resolution. A remarkable step forward in the enhancement of spectroscopic lateral resolution was obtained when tip-enhanced Raman spectroscopy (TERS) was proposed [13,14,15]. TERS measurements are generally demanding and not always versatile when coupled with an EC cell, where electrolytes at varying pHs are employed. Tips can be partially damaged by strong acids, oxide layers can form in alkaline solutions, and metal corrosion is possible at different pH values depending on the metal employed for the tip coating. Moreover, TERS data interpretation is not always straightforward, and TERS still shows some issues, as recently reported by Britz-Grell [16]. Our work proposes an alternative approach, at the expense of the extreme spatial resolution of TERS, where a Raman and an AFM system are coupled to collect both topographic and spectroscopic information during CV.

Here, we describe an innovative and versatile set-up that couples Raman spectroscopy and AFM inside an EC cell for studying electrode surface properties in operando conditions. Final outputs of the set-up include: (i) topographic images (AFM), (ii) chemical maps (Raman), and (iii) electrochemical characterization (e.g., cyclic voltammetry, CV). These are acquired in parallel during the chemical reactions. The Raman laser beam can be focused very close to the AFM tip (distance of about 1 μm), with stable and reliable positioning. Therefore, spatial topography and chemical maps can be acquired simultaneously on the studied sample area. Truthfully, an instrument similar to the one here discussed has been recently developed and used to analyze phenomena at the solid–liquid interface [17]. Nonetheless, our apparatus has substantial differences: (1) the instrument is fully commercial; (2) the system can study any kind of sample; (3) more than a laser source can be employed in the Raman spectroscopy measures; (4) this instrument is able of working in both contact and non-contact mode so that phase-contrast images can be collected.

We describe and show the potentiality of the new system through a targeted application: the spectroscopic and microscopic characterization of highly oriented pyrolytic graphite (HOPG) during anion intercalation in acidic media. Originally, the relevance of this topic rapidly increased after the results obtained by Foley (enhancement conductivity of the intercalated HOPG [18]) and Besenhard (oxidized HOPG as an electrochemical storage system [19]). The anion intercalation model in HOPG was developed by Goss and co-workers in the early 1990s [20]. There is currently a renewed interest in evaluating the applicability of graphite electrodes in commercial devices such as ion transfer batteries, which rely on the electrochemical intercalation mechanism [21,22]. When HOPG is employed as an anode inside an acid environment (such as diluted sulfuric acid), solvated anions intercalate inside the stratified structure of HOPG [20] according to the following symbolic equation: (1)$$C_{x}+A^{-}+y(\mathrm{solv})\;⇆[C_{x^{+}}\;A^{-}]y(\mathrm{solv})\;+\;e^{-}$$

A reaction of a certain amount of carbon atoms (C*_x_*) with the solvated anions (A^−^) occurs at any buried layer where the intercalation process takes place. Because of the solvated anion intercalation and the application of EC potentials above the oxygen evolution reaction, some gases are produced (namely CO, CO_2_ and O_2_) [23]. Therefore, the graphite basal plane is deformed by the formation of blisters, which characterize the surface of intercalated graphite [20]. Spectroscopic studies (e.g., Raman spectroscopy) have been also conducted [24,25] for a better understanding of the properties of intercalated graphite. These studies show the evolution of new features in the Raman spectrum during the intercalation process. Alongside the characteristic G peak (1581 cm^−1^) of the pristine sample, two new features evolve during the intercalation process. The formation of defects causes enhancement in the D peak (1360 cm^−1^) intensity while a second feature evolves, the G_i_ peak (1604 cm^−1^), that is related to the intercalation of anions. Although these observations and the relation between peaks and morphology have been discussed for a long time, the precise correlation between the Raman spectrum features and the topographic evolution of HOPG via AFM is still unclear. Unprecedented insights can be offered by the innovative experimental set-up designed by NT-MDT. We discuss in Section 3 why some key information can be obtained only if both spectroscopy and microscopy are coupled together with EC techniques. 

## 2. Materials and Methods

The new NTEGRA Spectra system commercialized by NT-MDT can collect reflected Rayleigh and Raman signals from the same side where the AFM scanner operates, which clearly allows for the acquisition of Raman data on opaque surfaces. NT-MDT engineered the adaptation of the Raman-AFM system to a real EC cell that could be filled with strong electrolytes and not just weak physiological solutions. The EC cell is driven by a (bi)potentiostat, which is useful for EC-STM measurements. The NT-MDT microscope is a sample scanning AFM, where the tip is kept fixed and the sample is positioned over a piezo-scanner that moves during the image acquisition. Sample scanning is crucial when Raman spectroscopy is added to AFM measurements: the impinging laser beam can be left in the same position during the sample scanning, which allows for the measurement of the same region of the sample that is probed using AFM. To optimize the superposition between the Raman map and the AFM image, a special tip (VIT_P) configuration is preferred (Figure 1b) to the usual one (Figure 1a); it is thus possible to focus the Raman laser beam adjacent to the tip apex. 

A confocal micro-Raman spectrometer is used for the acquisition of the Raman spectrum. Different laser sources can be employed (i.e., 405 nm, 473 nm, 532 nm, 633 nm, with a maximum power of 30 mW). Spectra documented in this work were acquired with an excitation laser of 532 nm, which yielded a high-quality assessment of the ion intercalation features in the Raman spectra. The NTEGRA Spectra system can focus the laser beam and collect the scattered light in different geometries. Among these, the axial geometry, where the objective is placed on the same side of the AFM tip, is particularly suitable for coupling with the EC cell, as depicted in Figure 1c. A system of screws allows a correct positioning of the objective above the tip holder, where the AFM tip is mounted. The coupled system is depicted in Figure 2a.

When the objective and the tip holder are placed inside an EC cell, the main two constraints are related to (i) the diameter of the EC cell, which is wide enough to allow the insertion of the Raman/AFM head and (ii) the overall small dimension of the EC cell, which is required to avoid a weight excess that can preclude the proper movement of the piezo-scanners. A technical solution for a compromise between these two constraints is depicted in Figure 2b. The sample (working electrode, WE) is connected backward, while the counter electrode (CE, a Pt wire) is bent to turn around the cell (not visible in Figure 2b). We employed a second Pt wire as a reference electrode (RE). While this is not a standard reference (PtQRef) because it does not provide a redox couple, it is stable (within ten mV) in an acidic electrolyte and shows a constant shift of about 740 mV with respect to the standard hydrogen electrode [26]. The EC cell is driven by a potentiostat coupled with the system. 

The employed HOPG sample inserted in the EC-cell is clearly visible in Figure 2b. The sample (grade ZYH) was provided by Optigraph and it was exfoliated using adhesive tape before each experiment. A diluted 1 M H_2_SO_4_ electrolyte was prepared for the experiment. The solution was purified by bubbling argon for several hours. The mechanical system depicted in Figure 2a was placed vertically above the EC cell. 

The Raman spectra were collected with a 532 nm laser source and a 600 lines/mm grating. The AFM topography was acquired in non-contact mode by exploiting VIT_P/IR (Tips Nano) tips (force constant = 60 N/m; resonance frequency in air 300 kHz, in liquid about 150 kHz) with a Au coating to ensure stability in acid environments. 

In the following, we present a selection of the data acquired during the experimental campaign. This required more than 20 samples, and the results were analyzed and compared to ensure the stability of the whole set-up.

## 3. Results and Discussion

Figure 3a depicts an image acquired upon immersion of the graphite sample in the electrolyte. This image is composed of the topography (upper side) and the AFM phase-contrast (lower part). The HOPG morphology shows the well-known (mono-) multi-atomic steps that separate two near terraces. A profile of a step, taken along the white dashed line, is superimposed on the image. The phase-contrast signal is almost uniform. This suggests a substantial uniformity of the pristine HOPG surface despite the fact that the measurements were acquired with the sample immersed in electrolyte.

The Raman spectrum (Figure 3b) was acquired on different regions inside the same area of the sample. Here, the spectrum is an average of several spectra characteristic of graphite before the electrochemical treatment. The spectrum is dominated by the intense peak (highlighted red region) located at 1582 cm^−1^ with a FWHM of 20 cm^−1^. This peak corresponds to the Raman active phonon with E_2g_ representation, found at the **Γ** point of the Brillouin zone (BZ) of the graphite crystal, and it is commonly referred to as the “G band” [27]. The so-called 2D band is visible at around 2700 cm^−1^ [27,28]. This second-order feature composed of several bands is due to two quanta transitions that involve the phonons responsible for the first-order D band. While the 2D band is always present in graphite spectra, in large, defect-free graphite crystals the D band has vanishing intensity because it is associated with phonons close to the **K** point of the BZ, which are inactive according to the symmetry selection rules of a perfect crystal. The label “D” stands for the word “defects”; indeed, the D band appears in microcrystalline graphites, namely in the presence of crystal edges and when structural or chemical defects break the crystal symmetry, thus allowing the activation of a peculiar double resonance mechanism [28]. Other features related to defectivity may appear in the spectrum, as in the D’ peak described below. In addition to the very strong 2D features, the second-order Raman spectrum of graphite may show other, very weak bands [28], such as the one at about 2450 cm^−1^ present in our spectrum. This band has received different assignments in the literature [28,29]. Near 1000 cm^−1^, we observe characteristic Raman features assigned to the sulfate and bisulfate ions [30,31].

Raman spectroscopy can be also coupled with the AFM during its scan. In this case, instead of giving a mean spectrum as above, a full Raman spectrum can be acquired within subsequent *regions* (each of them having a size of about 0.25 μm^2^) along each line scanned by the AFM tip. Consequently, we can imagine the original full (20 × 20) μm^2^ scanned area as composed by (0.25 × 0.25) μm^2^ tiles where a full Raman spectrum is acquired. For each tile (or region), it is possible to select one or more Raman spectral features and assess their signal intensity in each acquired *region* (e.g., the G peak highlighted in panel b). The final result is a sort of chemical (Raman) map, as shown in the following. 

The anion intercalation inside the graphite can be driven by electrochemistry. Figure 4a depicts the current density versus a Pt electrode (PtQRef) during the very first CV. The latter was acquired with starting and maximum EC potentials of 300 mV and 1300 mV, respectively. The selected scan rate was 25 mV/s, consistent with typical practice [32]. The oxygen evolution potential (OEP) was at about 800 mV. Here, we observe a significant enhancement of the faradaic current. When the EC potential was reversed, the flowing current shows a wide negative (cathodic) peak. The latter is usually interpreted in terms of a de-intercalation process [20,22,32]. 

As already discussed in a previous work [33], the intensity of the defect-related Raman D peak starts increasing when the EC potential is close to the OEP (Figure 4b). It is then stable during the anodic potential sweep. After reversing the EC potential, the intensity of the D peak starts to increase again when the potential is about (800 ± 100) mV (Figure 4b), i.e., where the CV (Figure 4a) displays a minimum in the faradaic current. One can also observe the intensity of the G peak shoulder (the so-called D’ at ~1620 cm^−1^) that shows an evolution similar to that of the D peak. More precisely, the D’ feature is assigned to CC stretching vibrational modes localized on the edges of the graphite layers [28], thus confirming the occurrence of carbon dissolution processes, with the formation of new, disordered edges on the most superficial layers [34].

Chemical (Raman) maps (see above) can be represented as a function of the applied EC potential during CV. All the reported intensity maps show normalized data with respect to the G peak intensity. Figure 5a shows the appearance of the D peak over an image of (20 × 20) μm^2^ as soon as the OEP is reached at about 800 mV. 

The normalized Raman map of the D peak, acquired at constant EC potential (300 mV) after CV (panel b), confirms a typically reported interpretation [33]. In this context, the D-peak is associated with the presence of defects on the crystal surface. Generally, the intensity of the D peak with respect to the G peak is no longer zero and increases over the entire analyzed surface, with some variations that can depend on the concentration of defects on the surface. In particular, Figure 5b shows a clear enhancement in D peak intensity in the region where AFM observes a clearly defective region (Figure 5c), probably caused by the significant carbon dissolution observed on the HOPG surface when high anodic potentials are applied [33].

In addition, the 2D band changes during the CV sweep. In particular, the intensity of the high frequency component (about 2719 cm^−1^) decreases during the CV sweep. From Figure 5b, we observe a reduction in this component when we reach the faradaic current enhancement. Apparently, in the electrode region where the D peak intensity shows a maximum, the 2D peak displays an intensity decrease (see panel d).

The possibility to collect topographic images under different acquisition modes offers important potentialities in the interpretation of the processes involved during the electrochemical reactions. By exploiting the non-contact mode, we can compare chemical maps and phase-contrast and morphological images. As an example, here we report the correlation between the G_i_ peak (located at around 1605 cm^−1^–1610 cm^−1^ [35] and characteristic of intercalated HOPG [32]) and some special areas that are observed in the phase-contrast images after the exposure of the graphite electrode to a relatively intense laser beam. Figure 6, acquired at a constant potential of 300 mV after CV, depicts a comparison between the map of the G_i_ peak intensity (a), the AFM phase-contrast image (b), the D peak map (c), and the electrode surface morphology (d). In panel (e), we report the Raman spectrum as collected in the brighter regions of the phase-contrast image. 

Thanks to the comparison between the chemical map and the phase-contrast information, it is possible to identify the regions (spatial localization) that mainly contribute to Gi peak intensity. The significance of this result is even more evident when comparing the Gi peak map with that of the D peak and/or the AFM morphology, in which case (Figure 6c, d, respectively) we clearly see that the Gi peak is not necessarily related to defect regions (as highlighted by the D map) even if it appears after the intercalation process. On the other hand, standard AFM morphology (panel d) is completely blind to this complex phenomenology and the graphite electrode wrongly appears unperturbed. 

## 4. Conclusions

Our study documents and discusses the potentiality of a new commercial experimental set-up able to combine AFM, Raman spectroscopy, and electrochemistry for joint analyses of electrodes in operando conditions. The new system is provided by NT-MDT (NTEGRA Spectra). It exploits a new mechanical head with the ability of (i) driving the laser beam of the Raman set-up very close to the AFM tip apex and (ii) collecting the reflected light in an axial configuration. This configuration is suitable for coupling with an EC cell, as the optical objective and the AFM head can be immersed inside the electrolyte. All metallic parts are screened to avoid interferences with electrochemical characterization (e.g., cyclic voltammetry). This experimental configuration is particularly suitable for studying reversible or irreversible processes. In such cases, with ordinary set-ups where electrochemistry, AFM, and Raman spectroscopy cannot all be combined, it is not possible to collect both microscopic and spectroscopic data on the same sample and experiments must be repeated on different specimens.

To demonstrate the potential of the new set-up, we studied the initial stages of anion intercalation of HOPG, when graphite is immersed in an acidic electrolyte. The insertion of molecules inside a stratified crystal represents a longstanding research topic that is well-studied by scanning probe techniques and spectroscopies (Raman included). Otherwise, published results are not always directly related to one another because the employed techniques require different conditions for a proper analysis. For example, the G_i_ Raman feature, which is characteristic of massive intercalated graphite, is observed after EC processes that could significantly affect the HOPG basal plane, hampering a microscopic investigation at the molecular-length scale. In such conditions, a correlation between the microscopic and spectroscopic data can be only speculative. Our innovative set-up overcomes such issues and offers the possibility of acquiring both AFM and micro-Raman data at the same time and in real time during EC processes.

However, some limitations must be considered. The AFM detection is based on the alignment of a laser beam on the back side of the cantilever. If the electrolyte is not transparent or turbid, the reflected laser beam is not able to reach the four-quadrant detector. The Raman/AFM scanner head is more massive with respect to a usual one because it is composed of an optical objective, a tip-holder, a piezo-system for non-contact measurements, etc. Consequently, if all these parts are not properly protected, some damage can occur when the system is used in strongly acidic or basic electrolytes. In addition, the employed material used to screen metallic parts (e.g., parafilm) cannot be too thick so as to avoid influencing the AFM scan.

We are strongly convinced that the NTEGRA Spectra represents a significant step forward to the correct and complete interpretation of EC processes that are at the center of promising research efforts in catalysis, Li-ion-batteries, corrosion phenomena, or geomaterials.

## Figures and Tables

**Figure 1 materials-16-02239-f001:**
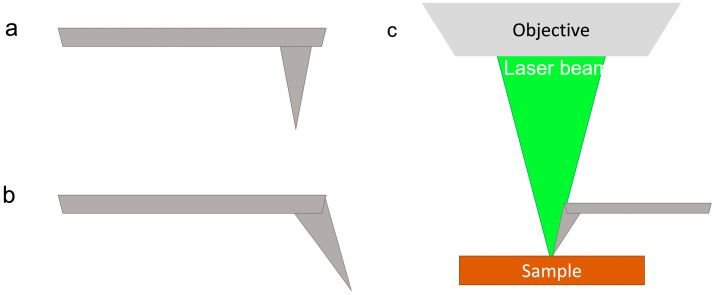
Sketch of (**a**) the usual tip geometry, (**b**) the special tip geometry optimized to acquire the Raman-AFM combined measurement; (**c**) axial coupling between the Raman system and the AFM set up, which highlights the special tip.

**Figure 2 materials-16-02239-f002:**
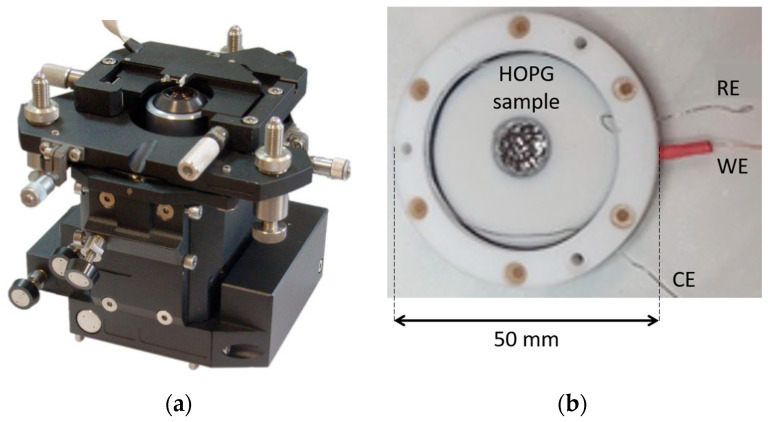
(**a**) An upside-down image (for a better view) of the mechanical system used for the laser beam positioning close to the AFM tip. (**b**)The EC-cell employed with the NTEGRA system. Electrodes (denoted as RE, WE and CE) and the HOPG sample are clearly visible in the photo.

**Figure 3 materials-16-02239-f003:**
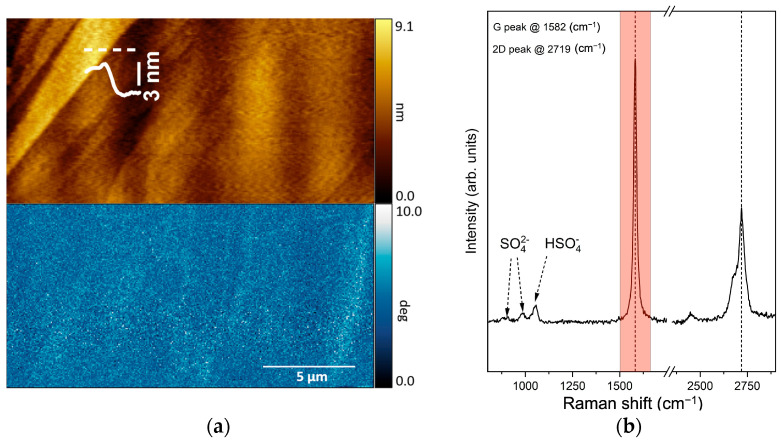
(**a**) A (20 × 20) μm^2^ AFM topography (upper panel) and the phase-contrast image (lower panel) of the HOPG electrode immersed in 1 M H_2_SO_4_ electrolyte with the sample biased at 300 mV with respect to the reference electrode (see the text for more details). (**b**) Raman spectrum acquired in a point inside the scanned area reported in panel a. The red region highlights the G peak feature.

**Figure 4 materials-16-02239-f004:**
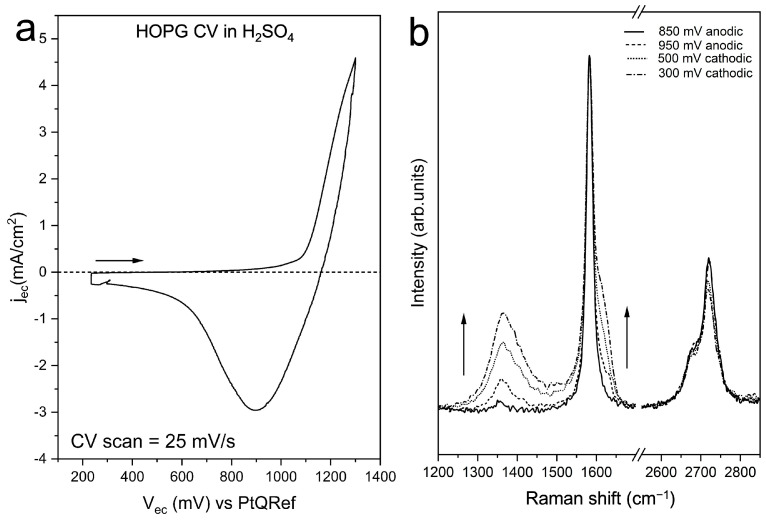
(**a**) A characteristic CV of the HOPG intercalation in diluted sulfuric acid electrolyte. (**b**) D peak evolution as a function of the applied potential. ”D”, “D2’”, “G” are characteristic labels of Raman features (see text for details).

**Figure 5 materials-16-02239-f005:**
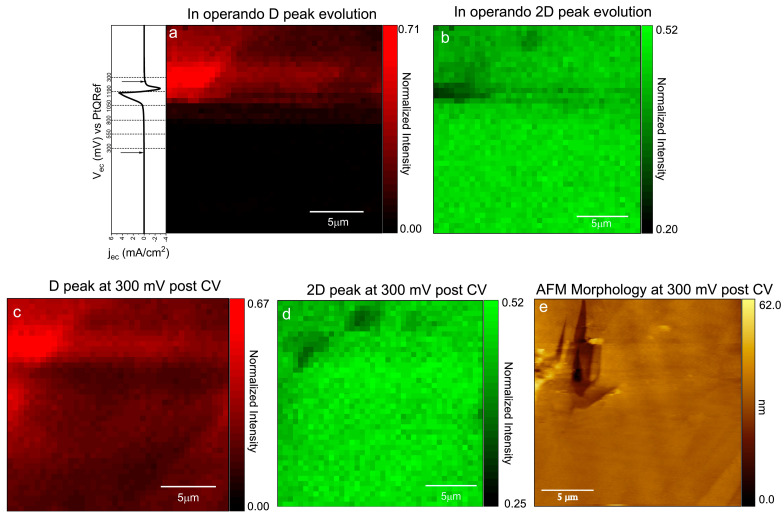
A (20 × 20) μm^2^ D peak (**a**) and 2D (**b**) intensity map acquired during the EC potential sweep (CV reported on the left side of the image); (**c**) D peak and 2D (**d**) intensity map acquired at constant 300 mV after the CV and compared to the AFM topography; (**e**) acquired in parallel to the Raman spectra. All the Raman maps show normalized data with respect to the G peak intensity.

**Figure 6 materials-16-02239-f006:**
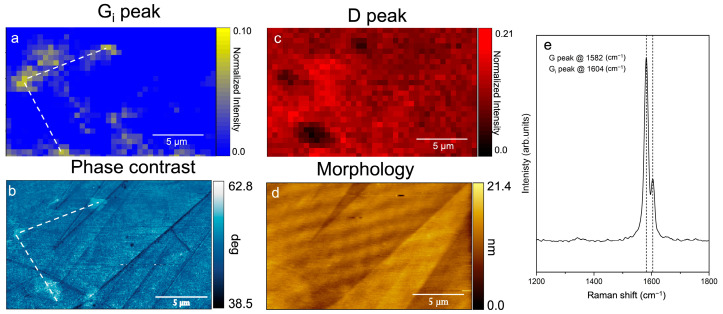
A (12 × 20) μm^2^ G_i_ intensity peak map (**a**) compared to the AFM phase-contrast image (**b**), the D intensity peak map (**c**) and the AFM morphology (**d**) acquired simultaneously in situ and in the same sample area. The Raman spectrum (**e**) is collected on the brighter areas of (**b**). All the Raman maps show normalized data with respect to the G peak intensity.

## Data Availability

The data are available by writing to the corresponding author.

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
