# Peer review of "A Combined Raman Spectroscopy and Atomic Force Microscopy System for In Situ and Real-Time Measures in Electrochemical Cells"

_materials, 2023, doi:10.3390/ma16062239_

Round 1

Reviewer 1 Report

This article presents a setup for the simultaneous characterization of Raman spectroscopy and atomic force microscopy to evaluate the electrochemical behavior of samples. However, more experimental samples need to be analyzed and, at the same time, compared and evaluated with the results provided through similar methods. In addition, the advantages and weaknesses of the current method should be investigated with similar methods.

Reviewer 2 Report

 During the reviewing process some recommendations and questions appeared (see below). 

-          line 74 – The dot should be after reference.

-          Equations (1.1) are not clearly understandable. The cation (anion) stays intact after electrochemical processes. Intercalation process should be denoted with subscript since it takes place.

-          line 165 – several hours is quite long period for argon purging. What is the necessity of such duration?

-          The legend of Figure 4a contains incorrect spelling cyclic voltammetry or cyclic voltammogram.

-          Figure 5a (CV part) should be more visible.

-          Figure 5c shows big region of defected surface. Very different D peak intensities (from one of the darkest to brightest sites) correspond to this region in Figure 5b. Authors should explain these results as well as provide more examples of revealed regularity (maybe in Supplementary materials).

-          What is the reason to apply the certain (300mV) constant potential after CV since all the intercalation processes caused by OER have already taken place?

-          Authors should reveal the expected limitations of approach in Conclusions. As example, what about the applicability of approach in the investigation of colored solutions or formation of polymeric films using electropolymerization et al.?

Reviewer 3 Report

This paper deals with the demonstration of an experimental setup. This setup is an insitu electrochemical cell allowing to couple electrical measurements with micro Raman spectroscopy and AFM. To demonstrate that the technique works, the authors studied the initial stages of anion intercalation in graphite. The paper is essentially technical to demonstrate the possibilities of a commercial set up. The results sound convincing. However, I find the scientific part (related to the analysis of the intercalation) not detailed enough. The results only correlate things without any clear understanding. I can accept this work, which is “setup oriented” but there are some comments below that should be addressed before.

In the introduction:

-          about TERS, it could be interesting to introduce/comment that two works which performed Tip enhanced Raman spectroscopy (AFM+Raman) in an electrochemical environment, and to position your work related to that ones:

Touzalin, Thomas. Et al

 Capturing electrochemical transformations by tip-enhanced Raman spectroscopy.

Current opinion in electrochemistry. (6)1. p.46 - 52. 10.1016/j.coelec.2017.10.016

Touzalin, Thomas. Et al.

Complex Electron Transfer Pathway at a Microelectrode Captured by in Situ Nanospectroscopy. Analytical chemistry (89)17. p.8974 - 8980. 10.1021/acs.analchem.7b01542

- Line 70-71: how much is that “close” ? One micron? More?

In the materials and method part:

-          is the word “place” at line 129 at the good place?

-          Figure6a line 143 has to be changed to figure 2a. Same comment line 157 about figure 6b and line 162 with figure 6, line 166 etc.

In the results part:

-          Line 179: “immersed”

-          Is there something interesting to show in the 2D region? I guess that intercalation will change the shape of the structured 2D band into a broader, less intense mono lorentzian. Can this information be used correlatively with that Gi band intensity?

-          A reference could be added to prove the bands are attributed correctly to SO42-. May be you will find interesting ones here:

Prieto-Taboada, Nagore et al.

The Raman spectra of the Na 2 SO 4 -K 2 SO 4 system: Applicability to soluble salts studies in built heritage.

Journal of Raman spectroscopy. (50)2. p.175 - 183. 10.1002/jrs.5550

-          Line 224 and related paragraph: is the D band raw data which is plot? What about the G band raw intensity? It could be interesting to show it as well. One could imagine that the focus has changed and the D band enhancement is only geometrical/optical. One way to better show this is to plot the ID/IG ratio, like it is done in the figure 4b.

-          G and D bands origin are not really presented. I only found the reference 25 about Ferrari’s seminal work, but it is not in the text. You should first use this reference. Also note that the field has evolved since 2007. You could find an update here (and see some references inside):

Merlen, Alexandre et al.

A Guide to and Review of the Use of Multiwavelength Raman Spectroscopy for Characterizing Defective Aromatic Carbon Solids: from Graphene to Amorphous Carbons.

Coatings  (7)10. p.153. 10.3390/coatings7100153

-          One question about the Gi band: you cite papers that give some physical origin but do not discuss this in details. You also cite the D’ band. It is not clear: do you argue that the D’ band is in fact this Gi band? I think this is not and the best way to investigate this is to do extra measurements at other wavelength and measure how the D’ band evolves with wavelength. It could be the same. In that case this is known that the D’/D ratio could depend on the kind of defect (vacancy,…).

-          Another question: I found this review paper that uses G1 and G2 bands. Could you comment about the nomenclature and, if useful, integrate it in your bibliography:

Rimkute, Gintare et al.

Synthesis and Characterization of Graphite Intercalation Compounds with Sulfuric Acid.

Crystals  (12)3. p.421. 10.3390/cryst12030421

               Final remark: why phase contrast and Gi band intensity correlate? Do you have any idea to suggest? Could it be related to the way layers of graphene interact together and the way this interaction is modified by the intercalation? It could be good to imagine an explanation at that step of the paper. One guess: I bet the 2D band shape changes in that region where the Gi band appears? If yes, you have here other information to show that interaction between layers is modified. If not, you have the contrary argument.

Round 2

Reviewer 1 Report

 Accept

Reviewer 2 Report

I find the author's reply satisfactory. The revised version of manuscript is appropriate to publication.

Reviewer 3 Report

Dear authors

I gave on the first report a major revision status, if you decided to do extra experiments/or do more data interpretation to feed the points I suggested. However, you did not do extra measurements or not extensive data interpretation but I understand that the main aim of the paper is not exactly related to the details I suggested. For the other points, your replys were nice.

Then, I can accept this paper, even if I am a bit disappointed by the fact you did not investigate more some points.

I hope you will do another deeper work

Regards